# Contrast Associated Acute Kidney Injury and Mortality in Older Adults with Acute Coronary Syndrome: A Pooled Analysis of the FRASER and HULK Studies

**DOI:** 10.3390/jcm10102151

**Published:** 2021-05-16

**Authors:** Rita Pavasini, Matteo Tebaldi, Giulia Bugani, Elisabetta Tonet, Roberta Campana, Paolo Cimaglia, Elisa Maietti, Giovanni Grazzi, Graziella Pompei, Gioele Fabbri, Alessio Fiorio, Andrea Rubboli, Gianni Mazzoni, Francesco Vitali, Matteo Serenelli, Gianluca Campo, Simone Biscaglia

**Affiliations:** 1Cardiovascular Institute, Azienda Ospedaliero-Universitaria di Ferrara, 44124 Ferrara, Italy; tblmtt@unife.it (M.T.); tonet.elisabetta@gmail.com (E.T.); robertacampana22@gmail.com (R.C.); graziella.pompei@outlook.it (G.P.); gioele.fabbri2@gmail.com (G.F.); alessiofiorio@virgilio.it (A.F.); Francesco.vitali90@gmail.com (F.V.); matteoserenelli@gmail.com (M.S.); cmpglc@unife.it (G.C.); simone.biscaglia@gmail.com (S.B.); 2U.O.C. Cardiologia, Ospedale Maggiore, Largo Nigrisoli 2, 40133 Bologna, Italy; giulia.bugani@gmail.com; 3Maria Cecilia Hospital, GVM Care & Research, 48033 Cotignola, Italy; paolocimaglia88@gmail.com; 4Department of Medical Science, University of Ferrara, 44121 Ferrara, Italy; e.maietti@ospfe.it; 5Department of Biomedical and Neuromotor Sciences, University of Bologna, 40127 Bologna, Italy; 6Center of Sport and Exercise Sciences, University of Ferrara, 44121 Ferrara, Italy; giovanni.grazzi@unife.it; 7Division of Cardiology, S. Maria Delle Croci Hospital, 48121 Ravenna, Italy; andrea.rubboli@auslromagna.it; 8Public Health Department, Azienda USL di Ferrara, 44121 Ferrara, Italy; mzo@unife.it

**Keywords:** acute coronary syndrome, percutaneous coronary intervention, contrast induced acute kidney injury, older patients, mortality, physical performance

## Abstract

Whether contrast-associated acute kidney injury (CA-AKI) is only a bystander or a risk factor for mortality in older patients undergoing percutaneous coronary intervention (PCI) is not well understood. Data from FRASER (NCT02386124) and HULK (NCT03021044) studies have been analysed. All patients enrolled underwent coronary angiography. The occurrence of CA-AKI was defined based on KDIGO criteria. The primary outcome of the study was to test the relation between CA-AKI and 3-month mortality. Overall, 870 older ACS adults were included in the analysis (mean age 78 ± 5 years; 28% females). CA-AKI occurred in 136 (16%) patients. At 3 months, 13 (9.6%) patients with CA-AKI died as compared with 13 (1.8%) without it (*p* < 0.001). At multivariable analysis, CA-AKI emerged as independent predictor of 3-month mortality (HR 3.51, 95%CI 1.05–7.01). After 3 months, renal function returned to the baseline value in 78 (63%) with CA-AKI. Those without recovered renal function (*n* = 45, 37%) showed an increased risk of mortality as compared to recovered renal function and no CA-AKI subgroups (HR 2.01, 95%CI 1.55–2.59, *p* = 0.009 and HR 2.71, 95%CI 1.45–5.89, *p* < 0.001, respectively). In conclusion, CA-AKI occurs in a not negligible portion of older MI patients undergoing invasive strategy and it is associated with short-term mortality.

## 1. Introduction

Several studies documented a strong association between contrast-associated acute kidney injury (CA-AKI) and an increased risk of adverse events, including mortality, need for dialysis, and hospital readmission for heart failure [1,2,3,4,5,6]. At the same time, both advanced age and hospital admission for myocardial infarction (MI) are significant risk factors for the occurrence of CA-AKI [7,8]. Lower baseline creatinine clearance, lower ejection fraction, high prevalence of heart failure at admission, and more complex coronary anatomy disease with severe calcifications are all common characteristics of older MI patients and all these factors further increase the risk of CA-AKI [9]. Thus, it is not surprising that nearly one out of five older adults undergoing cardiac catheterization for acute MI experience CA-AKI with associated increased short-term mortality [10]. However, recent evidence challenged the dogma that considers CA-AKI to be a direct cause of serious adverse events as mortality, need of dialysis, or persistent kidney impairment, suggesting that it could be only a marker (or a more frequent complication) of patients at higher risk for these complications [2,11,12,13]. Trying to better elucidate the relationship between age, CA-AKI and mortality, we analyzed the occurrence and prognostic implication of CA-AKI in two prospective studies focused on patients aged 70 years or more admitted to hospital for acute coronary syndrome (ACS) and undergoing invasive treatment. 

## 2. Materials and Methods

### 2.1. Study Design

The present analysis was carried out by using the data of the populations of two different studies. The first one was the “Frailty in Elderly Patients Receiving Cardiac Interventional Procedures” (FRASER) study (NCT02386124) [14,15,16]. In brief, this multicentre, observational, prospective study analysed the frailty status of 402 adults aged ≥ 70 years admitted to four Italian hospitals, with a diagnosis of ACS [14,15,16]. The second one was the “Physical Activity Intervention for Patients with Reduced Physical Performance After Acute Coronary Syndrome” (HULK) study (NCT03021044) [17,18,19]. In the latter randomized clinical trial, 485 adults aged ≥70 years admitted to three Italian hospitals for ACS were screened for physical performance at hospital discharge and then, if fulfilling inclusion and exclusion criteria, were randomized to health education or exercise intervention [17,18,19].

### 2.2. Study Measurements 

A large amount of clinical and management data, including demographics, previous medical history, comorbidities, echocardiographic and laboratory data, and treatments were collected in both studies. Starting from the individual dataset of both studies, a single database was generated under the supervision of the principal investigator (GC). Mehran risk score was not a prespecified variable available in the database. It is ad hoc calculated for the present analysis based on clinical and procedural variables available in the database [20]. The variables for Mehran risk score computation were age greater than 75 years, hypotension, congestive heart failure, intra-aortic balloon pump, serum creatinine, diabetes, anemia, and volume of contrast [20]. Left ventricular ejection fraction was calculated by the modified biplane Simpson method.

### 2.3. Procedures for CA-AKI Prevention

Indication for coronary artery angiography and percutaneous coronary intervention (PCI) were in keeping with international guidelines and institutional protocols [15,17]. Angiographies and PCI were performed by experienced operators in high-volume hub centers. The operators and centers shared protocols to minimize contrast media administration and the risk of CA-AKI. All patients received pre- and post-hydration. ST segment elevation myocardial infarction (STEMI) patients were treated with a saline infusion at a rate of 0.5 to 1 mL/kg per hour from admission and up to 24 h after the procedure. Saline infusion at a rate of 0.5 to 1 mL/kg per hour for 12 h before and up to 24 h after the procedure was administered to NSTEACS patients, or in the case of staged procedure. Automatic systems for contrast injection and low-osmolal contrast agent (Omnipaque 350, GE Healthcare, Chicago, IL, USA) were used in all centers. The general recommendation was to estimate the maximal allowable contrast dose (MACD), defined as: 5 mL × body weight (in kg)/serum creatinine (in mg/dL), with a maximum dose of 300 mL [21] and not to exceed it. If necessary, staged procedures (after at least 3 days) were suggested. 

### 2.4. Definition of CA-AKI 

The occurrence of CA-AKI was prospectively assessed. Serum creatinine (Beckman Coulter Inc., Brea, CA, USA) was measured at hospital admission (baseline value before any procedure) and every day until hospital discharge. Serum creatinine was expressed as mg/dl and creatine clearance was estimated by the Cockroft-Gault formula. CA-AKI was defined based on the Kidney Disease Improving Global Outcomes (KDIGO) criteria: increase in peak serum creatinine of ≥0.3 mg/dL from baseline or ≥1.5 times baseline within 48–72 h from the invasive procedure [22]. In patients with CA-AKI, serum creatine was measured again after 3 months [22]. If the 3-month value returned to baseline value (according KDIGO criteria) the patient was classified as recovered renal function. On the contrary, if the values persisted in being altered (according KDIGO criteria), the patients were classified as not having recovered renal function [22]. An independent central core-lab reviewed and adjudicated the occurrence of CA-AKI using original source documents. 

### 2.5. Clinical Follow-Up

All patients received follow-up visits at 1, 6, 12 months, and then yearly. Patients with CA-AKI received an additional 3-month visit with blood sample withdrawn for serum creatinine determination. Patients were examined and asked about the occurrence of adverse events [15,17]. Source documentation regarding each adverse event was collected [15,17]. The clinical events committee, whose members were unaware of the patients’ characteristics, centrally evaluated all events. Clinical follow-up was censored in April 2020. The FRASER study enrolled patients from December 2014 to October 2016, then a 3-year follow-up was available for all (median 1540 (1310–1700) days). On the contrary, the HULK study included patients from January 2017 to April 2018, then a 2-year follow-up was available for all (median 840 (705–1015) days). Overall, the median follow-up of the study population of the present analysis was 1023 (740–1446) days.

### 2.6. Outcomes 

The primary outcome of the present analysis was all-cause mortality. The first aim of the study was to test the relationship between CA-AKI and 3-month mortality. The second aim was to assess the relationship between recovered and not recovered renal function and all-cause mortality beyond the 3-month time point. The major predictors of CA-AKI were identified. 

### 2.7. Statistical Analysis

Continuous data were tested for normal distribution with the Kolmogorov–Smirnov test. Normally distributed values were presented as mean ± standard deviation (SD) and compared using *t*-test, otherwise median value (interquartile range, IQR) and the Mann–Whitney U test were used. Categorical variables were summarized in terms of counts and percentages and were compared by using the Fisher’s exact test. The estimation of the cumulative primary endpoint rate was found by using the Kaplan–Meier method, and groups were compared with the log-rank test. To better discriminate the effect of CA-AKI on short vs. long-term mortality, we performed analyses with the landmark set at 3 months. The 3-month time point was selected based on previous studies [23]. All variables included in Table 1 were tested using univariate Cox regression as predictors of adverse events. Variables showing a *p*-value < 0.05 were included in a multivariable model in association with CA-AKI with backward stepwise modelling approach. Variables remaining significant with a threshold *p*-value ≤ 0.05 were retained as final predictors. Cox regression analyses were performed for 3-month mortality and for mortality beyond the 3-month time point. The independent risk factors for CA-AKI were calculated using logistic regression. All tests were 2-sided, and the statistical significance was defined as *p* < 0.05. All analyses were performed with Stata 13 by the staff of the Center for Clinical Epidemiology of the School of Medicine at the University of Ferrara (Ferrara, Italy).

## 3. Results

Starting from 887 older ACS patients, we excluded 5 (0.6%) patients on chronic dialysis, 3 (0.3%) patients not undergoing PCI and 9 (1%) patients due to missing data. Thus, the final study population involved 870 (98%) older ACS adults undergoing PCI (Table 1). The mean age was 78 ± 5 years and 247 (28%) patients were females. Overall, 275 (32%) patients were admitted for STEMI. According to KDIGO definition, CA-AKI was observed in 136 (16%) patients. It was severe (stage 3 of AKI, defined as 3-time baseline creatinine, or serum creatinine > 4 mg/dL or initial of dialysis [22]) in 6 patients and moderate in 17 patients (stage 2 of AKI, defined as 2 to 2.9-time baseline creatinine [22]). Of note, three of the patients with severe AKI (2.2%) required dialysis during index hospitalization. Several baseline characteristics differed between patients with or without CA-AKI (Table 1). Patients with CA-AKI required a longer hospitalization stay (8 (7–10) days vs. 5 (4–6) days, *p* < 0.001). 

### 3.1. CA-AKI and 3-Month Mortality

At 3 months, all-cause mortality occurred in 26 (3%) patients. Thirteen (9.6%) patients with CA-AKI died as compared with 13 (1.8%) of those without (*p* < 0.001). The unadjusted cumulative occurrence of 3-month mortality was significantly different between subgroups, being higher in those with CA-AKI (*p* < 0.001) (Figure 1). At the univariate analysis, several clinical, laboratory, and angiographic variables, as well as CA-AKI, were associated with 3-month mortality (Table 2). At multivariable analysis, after correction for potential confounding factors, CA-AKI emerged as an independent predictor of 3-month death (HR 3.51, 95%CI 1.05–7.01) (Table 2).

### 3.2. Three-Month Recovery of Renal Functioned in Patients with CA-AKI

Out of 136 patients with CA-AKI, 123 reached the 3-month visit. Overall, 78 (63%) patients showed partial or complete recovery of renal function to the baseline level within 3 months (recovered renal function group). Renal function remained impaired in 45 (37%) patients (not recovered renal function group). Baseline characteristics of the two subgroups are shown in Table 1. Of note, 4 out of the 6 patients with severe CA-AKI and 3 out of 17 with moderate CA-AKI in the index hospitalization did not recover baseline renal function. Prior CABG, peripheral artery disease, and lower baseline creatinine clearance were more common in patients with not recovered CA-AKI (Table 1). In the same group, MACD was lower and the need of contrast media overcame MACD in a higher percentage of cases (Table 1).

### 3.3. CA-AKI and Mortality beyond 3-Month Visit

Overall, 844 patients reached the 3-month time point. Afterwards, 134 (16%) patients died. In particular, 96 (13%) patients belonged to the no CA-AKI group, 17 (22%) to the recovered renal function group and 21 (47%) to the not recovered renal function group (*p* < 0.001). Unadjusted cumulative mortality beyond 3 months was higher in patients with not recovered renal function (*p* = 0.008 vs. recovered renal function group and *p* < 0.001 vs. no CA-AKI group) (Figure 1). After correction for potential confounding factors, the not recovered renal function group remained associated with an increased risk of mortality as compared to recovered renal function group and no CA-AKI subgroups (HR 2.01, 95%CI 1.55–2.59, *p* = 0.009 and HR 2.71, 95%CI 1.45–5.89, *p* < 0.001, respectively). As compared to the no CA-AKI group, unadjusted cumulative mortality beyond the 3-month marker was higher in patients with recovered renal function (*p* = 0.043) (Figure 1). After correction for potential confounding factors, the difference was not significant (HR 1.07, 95%CI 0.61–1.91, *p* = 0.751) and patients of the recovered renal function group were not associated with a higher risk of mortality after the 3-month timepoint.

### 3.4. Predictors of CA-AKI

The analysis of predictors among baseline and procedural characteristics of CA-AKI was reported in Table 3. By multivariable analysis, baseline creatinine clearance, total amount of contrast media higher than the MACD, and Mehran risk score were the strongest predictors of CA-AKI (Table 3). 

## 4. Discussion

The main findings of the present analysis can be summarized as follows: In older adults admitted to hospital for MI and undergoing invasive strategy with PCI, CA-AKI is a frequent complication occurring in 16% (95%CI 13–18%) of patients,baseline creatinine clearance, total amount of contrast media higher than the MACD and Mehran risk score emerged as independent predictors of CA-AKI,the occurrence of CA-AKI was independently associated with 3-month mortality,Among patients with CA-AKI, renal function did not return to baseline values after 3 months in 37% (95%CI 28–46%), and these subjects showed a significantly higher risk of death.

The mean age of the population, and consequently the number of older adults admitted to hospital for MI, is increasing worldwide [16]. Older MI patients represents the subgroup at higher risk of death and early invasive strategy clearly showed its benefit in terms of the reduction of death and ischemic complications [16,18]. However, complications of invasive procedures tend to be frequent in older MI patients, and thus it is not surprising that advanced age is a well-established and strong risk factor for the occurrence of CA-AKI [10]. However, CA-AKI is a highly debated entity. Since the first reports in the 1950s [24], CA-AKI has been described both as a consequence of contrast media administration and as a silent, but deadly complication. In recent years, these assumptions have been challenged. The relationship between contrast administration and AKI is highly variable, strongly influenced by baseline risks, comorbidities, and other potential causes (i.e., atheromatous embolic debris) [25]. Similarly, some authors have suggested that CA-AKI is observed in subjects with several comorbidities, worse clinical presentation, complicated course of the disease, and then the increased mortality could be determined not by CA-AKI itself, but by the interaction of several factors of which CA-AKI is a bystander [11,12,13]. Our analysis is focused on older MI patients undergoing invasive strategy and PCI and represents an ideal subset of patients to try to clarify the missing pieces of the puzzle. Despite the inclusion of older MI patients, we did not find an excessively higher occurrence of CA-AKI as compared to previous studies on MI patients [26], and it was slightly lower as compared to that reported in the Comprehensive Evaluation of Risk Factors in Older Patients with AMI (SILVER-AMI) study (16% vs. 19%) [10]. 

We may suppose that these differences are explained by using modern contrast media and by the systematic application of hydration protocols. As in previous studies [8], we confirmed that patients experiencing CA-AKI have a worse short-term outcome. In addition, after correction for potential confounding factors, CA-AKI was an independent predictor of short-term death. This result further emphasizes the importance for physicians to minimize the occurrence of CA-AKI. As shown by previous trials [10], the first and most important step is the identification of patients at risk. The second step is the application of appropriate fluid administration. As suggested by current guidelines [27], hydration protocols were systematically applied, and we may hypothesize that they were able to reduce the burden of CA-AKI in our high-risk population. The third step is the minimization of contrast dose. Our analysis sheds new light on the importance of the MACD. We found that the relationship between contrast media and CA-AKI was not linear, but that contrast use is associated with CA-AKI when the MACD is exceeded. This finding deserves attention because it has several clinical implications. The total amount of contrast use is influenced by the experience of the operator and by the complexity and extension of the coronary anatomy requiring revascularization [25]. Complete revascularization should be considered the gold standard for MI patients [28]. However, evidence supporting this strategy has been mainly generated in patients with STEMI and a mean age of around 60 years [29]. In older patients with multivessel disease the potential benefit of a complete or extensive revascularization should be weighed against the risk of CA-AKI [30]. Operators should consider not to overcome MACD and eventually they should pursue a reasonable or physiology-guided coronary revascularization [30]. At the same time, it has to be noted that the risk profile of older MI patients with CA-AKI was significantly higher and more complex [26]. It is relevant to acknowledge that the design of the study (observational) is not enough to infer causality and to clearly demonstrate if CA-AKI is a direct or indirect determinant of death. 

It has been reported that more than 50% of MI patients with CA-AKI have a complete or partial recovery of the renal function [25]. We confirmed this finding in our population of older MI patients. Around 60% of subjects with CA-AKI showed a return to baseline values of renal function after 3 months. Interestingly, this recovery is associated with an important clinical benefit, because beyond the 3-month time point, the rate of adverse events is similar to the one of patients without CA-AKI. On the contrary, the minority of patients without recovery of the renal function showed a significantly higher long-term mortality. However, our analysis did not allow for discriminating from the beginning who will have recovered or not recovered CA-AKI. Then, waiting for future studies able to clarify this issue, physicians can only pursue any effort to avoid CA-AKI. 

### Study Limitations

Our study has some limitations. First, this is a post hoc analysis and we cannot exclude the presence of unmeasured confounders. Second, the recruitment of patients was performed in a limited number of cardiology units. Therefore, the generalizability of our findings requires confirmation in a larger scenario. Third, we re-assessed renal function at a single time point (3 months after acute event) and only in patients suffering from CA-AKI in the index hospitalization. Fourth, in both studies (FRASER and HULK) the enrollment of the patients was performed at discharge. This has a huge implication in the interpretation of data because: (i) patients with low physical performance and high burden of comorbidities (including renal failure) have been excluded; (ii) patients transferred to another hospital or to another ward were not included in the analysis. Next, considering that current guidelines recommend hydrating patients undergoing coronary angiography [27], in our center all patients are treated with hydration before and after coronary angiography, and this might have reduced the risk of CI-AKI development in the study population. 

## 5. Conclusions

Despite the use of modern contrast media and hydration protocols, CA-AKI occurs in a relevant portion of older MI patients undergoing invasive strategy and PCI. Patients with CA-AKI showed an increased risk of short-term death. Around two-thirds of the patients with CA-AKI recovered baseline renal function in the first 3 months, and their long-term prognosis was similar to that of patients without CA-AKI.

## Figures and Tables

**Figure 1 jcm-10-02151-f001:**
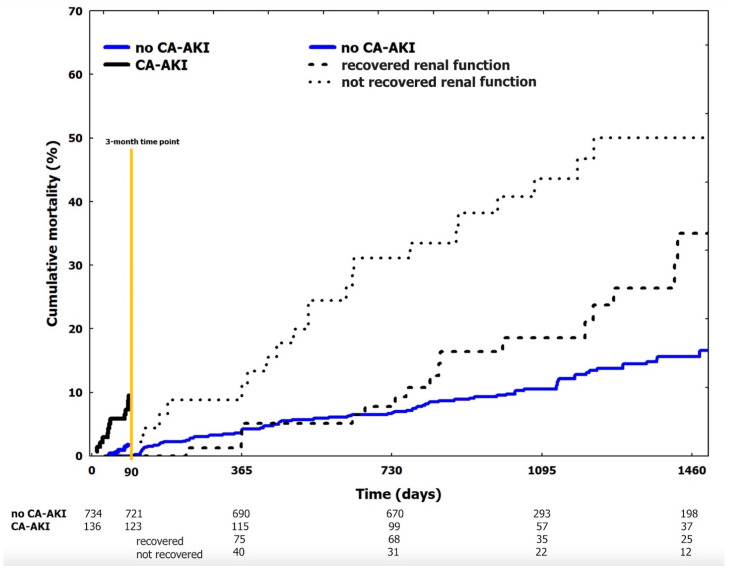
Cumulative occurrence of mortality (landmark analysis). Landmark analysis at 3 months. Continue blue line: no CA-AKI. Continue black line: CA-AKI. After the 3-month time point CA-AKI patients are stratified in those who recovered baseline renal function (dotted black line) vs. those who not recovered baseline renal function (pointed black line). CA-AKI: contrast associated acute kidney injury.

**Table 1 jcm-10-02151-t001:** Baseline characteristics.

	All (*n* = 870)			3-Month Renal Function in CA-AKI pts (*n* = 123)		
	No CA-AKI (*n* = 734)	Yes CA-AKI (*n* = 136)	*p*	Recovered (*n* = 78)	Not Recovered (*n* = 45)	*p*
Age, (years)	77 ± 5	80 ± 6	<0.001	79 ± 7	79 ± 5	0.454
Female sex, (%)	194 (26)	53 (39)	0.004	29 (37)	18 (40)	0.847
BMI, (Kg/m^2^)	26 (24–29)	27 (24–30)	0.219	27 (25–30)	27 (25–29)	0.843
**Medical history, no. (%)**						
Diabetes	221 (30)	41 (30)	0.926	24 (31)	13 (29)	0.987
Hypertension	599 (82)	125 (92)	0.005	72 (92)	40 (89)	0.528
Hyperlipidemia	391 (53)	54 (40)	0.006	35 (45)	17 (38)	0.456
Current smoker	153 (21)	44 (32)	0.005	26 (33)	18 (40)	0.558
MI	204 (28)	51 (37)	0.029	31 (39)	16 (35)	0.702
PCI	203 (28)	37 (27)	0.997	23 (29)	8 (18)	0.196
CABG	75 (10)	22 (16)	0.067	9 (11)	12 (16)	0.045
COPD	39 (5)	25 (18)	<0.001	17 (22)	6 (13)	0.338
PAD	166 (23)	46 (34)	0.007	17 (22)	21 (47)	0.004
**Presentation characteristics**						
STEMI, no. (%)	228 (31)	47 (34)	0.481	27 (35)	10 (22)	0.160
Killip class ≥ 2, no. (%)	65 (9)	25 (18)	0.002	13 (17)	7 (16)	0.901
SBP, mmHg	135 ± 30	133 ± 35	0.991	133 ± 35	134 ± 35	0.987
Heart rate, bpm	85 ± 20	89 ± 25	0.254	90 ± 25	88 ± 28	0.574
Hypotension *	30 (4)	10 (7)	0.148	4 (5)	4 (9)	0.663
Anemia ^†^	318 (43)	80 (59)	<0.001	42 (54)	27 (60)	0.635
IABP	5 (0.7)	4 (2.9)	<0.001	2 (2.5)	3 (6.6)	0.524
White blood cells, (u/μL)	8.8 (7.3–10.5)	10.9 (8.9–14.5)	<0.001	10.7 (8.1–13.2)	11.2 (8.9–12.9)	0.664
Hemoglobin, (g/dL)	12.5 (11–13.8)	11.2 (9.5–12.8)	<0.001	11.6 (9.8–13.1)	10.8 (9.4–12.7)	0.288
Albumin, (g/dL)	3.4 (3.2–3.7)	3.5 (3.1–3.7)	0.337	3.5 (3.2–3.7)	3.5 (3.3–3.7)	0.956
Creatinine Clearance, (mL/min)	59 (48–73)	44 (32–54)	<0.001	47 (40–61)	41 (28–48)	0.007
LVEF, (%)	50 ± 10	49 ± 12	0.708	53 ± 10	49 ± 11	0.088
Troponin I at peak, (ng/dL)	59 (25–84)	60 (28–85)	0.734	60 (30–90)	60 (25–88)	0.977
CK-MB at peak, (ng/dL)	31 (24–69)	35 (25–80)	0.481	35 (24–85)	35 (25–80)	0.892
**Procedural data**						
Radial access, no. (%)	705 (96)	130 (96)	0.998	76 (97)	43 (96)	0.879
Multivessel disease, no. (%)	558 (76)	102 (75)	0.883	61 (78)	30 (67)	0.201
Multivessel PCI, no. (%)	270 (37)	57 (42)	0.299	34 (57)	19 (47)	0.411
Staged procedure, no. (5)	110 (15)	19 (14)	0.436	10 (13)	6 (13)	0.895
MACD, (mL)	355 (299–423)	280 (223–330)	<0.001	293 (256–382)	267 (199–302)	0.005
Total contrast media, (mL)	170 (120–243)	179 (125–323)	0.968	179 (125–212)	200 (133–279)	0.144
Total contrast media > MACD, no. (%)	34 (5)	28 (20)	<0.001	5 (7)	18 (43)	<0.001
Mehran score, u.	8 (5–11)	10 (8–15)	<0.001	9 (6–13)	11 (8–15)	0.019
**Medical therapy at discharge**						
ACE-inhibitor	528 (72)	63 (46)	<0.001	37 (47)	23 (51)	0.694
Angiotensin receptor blocker	127 (17)	26 (19)	0.610	17 (22)	7 (16)	0.408
Beta-blocker	582 (79)	108 (79)	0.923	62 (79)	36 (80)	0.861
Statin	667 (91)	122 (90)	0.871	70 (90)	40 (89)	0.963
Diuretic	236 (32)	77 (57)	<0.001	38 (49)	28 (62)	0.148

CA-AKI: contrast associated acute kidney injury. BMI: body mass index. MI: myocardial infarction. PCI: percutaneous coronary intervention. CABG: coronary artery bypass graft. COPD: chronic obstructive pulmonary disease. PAD: peripheral artery disease. STEMI: ST-segment elevation myocardial infarction. SBP: systolic blood pressure. IABP: intra-aortic balloon pump. LVEF: left ventricular ejection fraction. MACD: maximum allowed contrast dose. ACE: angiotensin converting enzyme. *: hypotension is defined as a systolic blood pressure ≤ 90 mmHg. †: anemia was defined as hemoglobin < 13 g/dL for men and <12 g/dL for women.

**Table 2 jcm-10-02151-t002:** Univariate and multivariable Cox regression analysis for 3-month mortality.

	Univariate			Multivariable		
	HR	95%CI	*p*	HR	95%CI	*p*
Age	1.23	1.16–1.31	<0.001	1.20	1.08–1.34	0.004
Female sex	3.52	1.62–7.63	0.001			
BMI	1.01	0.91–1.35	0.841			
Diabetes	1.25	0.75–1.89	0.836			
Hypertension	1.07	0.71–1.63	0.740			
Hyperlipidemia	0.89	0.65–1.62	0.578			
Current smoker	1.04	0.73–1.48	0.827			
MI	0.71	0.28–1.76	0.713			
PCI	0.77	0.53–1.10	0.156			
CABG	0.98	0.60–1.63	0.950			
COPD	1.07	0.25–4.48	0.930			
PAD	2.71	1.26–5.85	0.011			
STEMI	6.03	2.54–14.2	<0.001			
Killip class ≥ 2	2.47	1.72–3.55	<0.001	2.21	1.08–2.84	0.032
SBP	0.99	0.97–1.03	0.652			
Heart rate	1.01	0.96–1.06	0.783			
Hypotension	1.82	0.76–4.54	0.541			
Anemia	1.96	1.44–2.68	<0.001	1.51	1.01–2.56	0.045
IABP	2.01	0.71–5.21	0.653			
White blood cells	1.07	1.02–1.12	0.008			
Hemoglobin	0.69	0.57–0.82	<0.001			
Albumin	0.17	0.06–0.48	<0.001			
Creatinine Clearance	0.94	0.92–0.96	<0.001			
LVEF	0.92	0.88–0.95	<0.001	0.89	0.83–0.95	<0.001
Troponin I at peak	1.01	0.98–1.12	0.496			
CK-MB at peak	1.01	0.89–1.12	0.189			
Radial access	0.91	0.78–1.76	0.654			
Multivessel disease	1.80	1.12–2.50	0.016			
Multivessel PCI	0.86	0.62–1.22	0.410			
ACE-inhibitor	0.81	0.59–1.09	0.148			
Angiotensin receptor blocker	0.56	0.38–1.19	0.311			
Beta-blocker	0.97	0.53–1.49	0.442			
Statin	0.91	0,64–1.78	0.645			
Diuretic	3.38	2.46–4.63	<0.001			
CA-AKI	5.65	2.63–12.1	<0.001	3.51	1.05–7.01	0.039

HR: hazard risk. BMI: body mass index. MI: myocardial infarction. PCI: percutaneous coronary intervention. CABG: coronary artery bypass graft. COPD: chronic obstructive pulmonary disease. PAD: peripheral artery disease. STEMI: ST-segment elevation myocardial infarction. SBP: systolic blood pressure. IABP: intra-aortic balloon pump. LVEF: left ventricular ejection fraction. ACE: angiotensin converting enzyme. CA-AKI: contrast associated acute kidney injury.

**Table 3 jcm-10-02151-t003:** Predictors of CA-AKI.

	Univariate Analysis			Multivariable Analysis		
	OR	95%CI	*p*	OR	95%CI	*p*
Age, (years)	1.08	1.04–1.11	<0.001	1.01	0.94–1.09	0.840
Female sex, (%)	1.78	1.21–2.60	0.003	1.62	1.01–2.63	0.049
BMI, (Kg/m^2^)	1.03	0.98–1.08	0.248			
**Medical history, no. (%)**						
Diabetes	1.01	0.67–1.49	0.992			
Hypertension	2.56	1.34–4.88	0.004	2.03	0.95–4.34	0.066
Hyperlipidemia	0.88	0.40–1.94	0.469			
Current smoker	1.51	0.72–1.71	0.504			
MI	1.56	1.06–2.29	0.023	1.22	0.74–2.02	0.438
PCI	0.98	0.65–1.47	0.914			
CABG	1.70	1.01–2.84	0.045	2.37	0.90–4.63	0.101
COPD	2.01	1.34–4.89	0.018	1.99	0.99–3.10	0.071
PAD	1.75	1.18–2.60	0.009	1.85	1.04–2.99	0.041
**Presentation characteristics**						
STEMI, no. (%)	1.17	0.80–1.73	0.421			
Killip class ≥ 2, no. (%)	1.63	1.24–2.15	<0.001	1.18	0.84–1.64	0.342
SBP, mmHg	1.01	0.85–1.26	0.985			
Heart rate, bpm	0.99	0.75–1.39	0.857			
White blood cells, (u/μL)	1.08	0.95–1.14	0.251			
Hemoglobin, (g/dL)	0.79	0.71–0.85	<0.001	0.94	0.84–1.06	0.310
Albumin, (g/dL)	1.27	0.73–2.19	0.398			
Creatinine Clearance, (mL/min)	0.96	0.95–0.97	<0.001	0.98	0.96–0.99	0.002
LVEF, (%)	0.99	0.98–1.02	0.707			
**Procedural data**						
Radial access, no. (%)	0.96	0.80–1.69	0.587			
Multivessel disease, no. (%)	0.95	0.62–1.45	0.798			
Multivessel PCI, no. (%)	0.98	0.65–1.48	0.943			
Staged procedure, no. (5)	0.78	0.46–1.33	0.366			
Total contrast media, (mL)	1	0.99–1.01	0.842			
Total contrast media > MACD, no. (%)	4.48	2.60–7.71	<0.001	2.32	1.18–4.57	0.015
Mehran score, u.	1.16	1.01–1.21	<0.001	1.07	1.01–1.13	0.022

CA-AKI: contrast associated acute kidney injury. OR: odds ratio. BMI: body mass index. MI: myocardial infarction. PCI: percutaneous coronary intervention. CABG: coronary artery bypass graft. COPD: chronic obstructive pulmonary disease. PAD: peripheral artery disease. STEMI: ST-segment elevation myocardial infarction. SBP: systolic blood pressure. LVEF: left ventricular ejection fraction. MACD: maximum allowed contrast dose.

## Data Availability

The data presented in this study are available on request from the corresponding author.

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
