# Peer review of "Contrast Associated Acute Kidney Injury and Mortality in Older Adults with Acute Coronary Syndrome: A Pooled Analysis of the FRASER and HULK Studies"

_jcm, 2021, doi:10.3390/jcm10102151_

Round 1
Reviewer 1 Report
Review for jcm-1182594
The relationship between CI-AKI and subsequent mortality was investigated in this observational study conducted in approximately 800 elderly patients undergoing angiography and potential PCI for ACS.
Comment #1
How was the Mehran risk score calculated? Some components are not described in the baseline table (e.g. use of hemodynamic support such as IABP and hypotension). The manuscript could be strengthened by adding these variables to the baseline table. Also, in addition to only hemoglobin as a continuous variable, anemia should be defined and reported as a dichotomous variable.
Comment #2
The authors should acknowledge a major limitation of their study which is the fact that all patients underwent pre- and posthydration. This is not routine clinical practice and emerging evidence suggests potential harm of hydration in patients with normal or only moderate renal function impairment.
Comment #3
Many, perhaps countless, prior papers have investigated the relationship between CI-AKI and mortality and the current analysis adds no meaningful new insights to the field. The relationship between persistent vs transient CI-AKI and mortality has also been investigated many times before. The small sample size is another important limitation.
Comment #4
Please add information on potentially nephrotoxic discharge medication (i.e. RAAS inhibitors, diuretics, NSAIDS etc)
Comment #5
Please add to the methods section how LVEF was defined.
Comment #6
Any data on peak biomarkers for myocardial injury in these ACS patients?
Author Response
Reviewer # 1
Question # 1
The relationship between CI-AKI and subsequent mortality was investigated in this observational study conducted in approximately 800 elderly patients undergoing angiography and potential PCI for ACS. How was the Mehran risk score calculated? Some components are not described in the baseline table (e.g. use of hemodynamic support such as IABP and hypotension). The manuscript could be strengthened by adding these variables to the baseline table. Also, in addition to only hemoglobin as a continuous variable, anemia should be defined and reported as a dichotomous variable.
Reply # 1
We are sorry for being not enough clear. We calculated the Mehran score based on clinical and procedural variables: age greater than 75 years, hypotension, congestive heart failure, intra-aortic balloon pump, serum creatinine, diabetes, anemia and volume of contrast. We amended the Table 1 adding the requested variables and we better specified in methods how the Mehran score was calculated.
Modified text: section Methods, page 2, lines 38-42
Mehran risk score was not a prespecified variable available in the database. It is ad hoc calculated for the present analysis based on clinical and procedural variables available in the database [20]. The variables for Mehran risk score computation were age greater than 75 years, hypotension, congestive heart failure, intra-aortic balloon pump, serum creatinine, diabetes, anemia and volume of contrast [20].
Modified text: Table 1
Added Reference number 20:
Mehran R, Aymong ED, Nikolsky E, et al. A simple risk score for prediction of contrast-induced nephropathy after percutaneous coronary intervention: development and initial validation. J Am Coll Cardiol 2004;44:1393e9.
Question # 2
The authors should acknowledge a major limitation of their study which is the fact that all patients underwent pre- and post-hydration. This is not routine clinical practice and emerging evidence suggests potential harm of hydration in patients with normal or only moderate renal function impairment.
Reply # 2
We really thank the Reviewer #1 for this comment. As suggested by current guidelines [Ref 26] the evaluation for the risk of developing contrast induced acute kidney injury is recommended as well as hydration for patients undergoing coronary angiography (class of evidence I, level C). Overall, the risk related to hydration is limited. The major risk is related to the worsening of heart failure, and this is the reason why our protocol in agreement with guidelines reduced fluid administration according to ejection fraction [Ref 25]. Anyway, we agree that a systematic hydration could limit the transferability of our data in larger scenario. We amended study limitation.
Modified text: section limitation of the study; page 11, lines 20-23
Next, considering that current guidelines recommend to hydrate patients undergoing coronary angiography [27], in our center all patients are treated with hydration before and after coronary angiography and this might have reduced the risk of CI-AKI development in the study population.
Question# 3
Many, perhaps countless, prior papers have investigated the relationship between CI-AKI and mortality and the current analysis adds no meaningful new insights to the field. The relationship between persistent vs transient CI-AKI and mortality has also been investigated many times before. The small sample size is another important limitation.
Reply #3
We thank the Reviewer for the precious comment. We agree that CI-AKI has been previously studied in several studies. However, we believe that our study has some peculiarities never considered before:
1) the focus on older patients (aged ≥ 70) all undergoing coronary angiography;
2) all patients received hydration (as correctly underlined by the Reviewer in the question before);
3) the description of the prognosis in agreement to recovery or not at 3-month of the renal function
We are aware that CA-AKI is a topic highly debated and investigated, but we believe that our data are helpful to add another piece of the puzzle allowing CA-AKI significance.
Question#4
Please add information on potentially nephrotoxic discharge medication (i.e., RAAS inhibitors, diuretics, NSAIDS, etc.)
Reply # 4
We thank the Reviewer for this important comment. In the previous version of the manuscript, we forgot to describe the medical treatment at hospital discharge. In the current version of the manuscript, we provided in Table 1 this information. Angiotensin converting enzyme inhibitors were, as expected, significantly less prescribed to patients with CA-AKI, who received significantly more diuretics. No differences were present in terms of angiotensin receptor blockers administered in the two groups. No patients received chronically NSAIDS in the cohort considered.
Modified text: Table 1
Question #5
Please add to the methods section how LVEF was defined.
Reply #5
We are sorry for the imprecision. The LVEF was calculated by the modified biplane Simpson method. The information is now available in the text.
Modified text: section Methods, page 2, lines 42-43
Left ventricular ejection fraction was calculated by the modified biplane Simpson method.
Question #6
Any data on peak biomarkers for myocardial injury in these ACS patients?
Reply # 6
Reviewer #1 is again right. The data is now reported in the Table 1.
Modified text: Table 1

Reviewer 2 Report
In this secondary analysis of two observational trials the authors assess the impact of contrast-associated AKI on long-term adverse outcomes. They report that CA-AKI is associated with increased 3months mortality and if persistent at 3 months also with long-term mortality.
General comment
- patients with ACS have many risk factors for AKI. The role of contrast in CA-AKI is being questioned. In this analysis the only “procedure-related” variable that shows a significant association with AKI is the dose in relation to MACD that by itself is highly correlated with baseline kidney function. An important risk factor for AKI in this population is hemodynamic instability: little data are given on this issue. Is the SBP and HR the value on admission or the worse value “around” the procedure? How many patients had hypotension and/or vasopressor or inotropic support?
- an association between AKI and long-term mortality has repeatedly been shown: findings of this analysis are therefore not novel
- For me the only conclusion is that patients that develop AKI when hospitalized for ACS have a higher 3months mortality, but the role of contrast in this is unclear. Whether the association between AKI and long-term mortality is causal cannot be inferred from this type of analyses.
Other comments
- the timing of CA-AKI is not mentioned – according to KDIGO the 0.3mg/dL increase should occur within 48h – CA-AKI is mostly defined as occurring with 3-5days
- no data are given on the severity of AKI, that may be an important determinant of “persistent AKI”
- the KDIGO definition indeed defines a 0,3mg/dL increase over 48h as stage 1 AKI. However, I am not sure that this 0,3mg/dL increase can be used to define “persistent” AKI after 3 months. I would suggest to limit the “persistent” AKI to the fold increase in creatinine
- in the AKI literature the term persistent AKI is used to define AKI that lasts for more than 48-72h (in contrast to transient AKI). AKI that persists after day 7 is called acute kidney disease, that is defined by fold change in creatinine (Chawla et al, Nature Nephrol 2017). After 3 months kidney function is defined by the CKD criteria. With regard to long-term kidney function the most important analysis would be the change in CKD status between admission and 3 months.
- is contrast the cause of “persistent” AKI? No data are available on events that may have caused AKI between the period of CA-AKI and 3 months (eg. how many patients were hospitalized or received RAAS blockers post-discharge)? How many patients without CA-AKI had “AKI criteria” at 3 months?
Statistics
- in the light of the “rule of ten” the n of variables in the logistic regression analyses is too high with respect to the n of events, especially for the 3 months mortality analysis where only 26 events were noted
- were variables checked for collinearity (that probably exists between eGFR and MACD)
Author Response
Reviewer # 2
Question#1
In this secondary analysis of two observational trials the authors assess the impact of contrast-associated AKI on long-term adverse outcomes. They report that CA-AKI is associated with increased 3 months mortality and if persistent at 3 months also with long-term mortality.
General comment
- patients with ACS have many risk factors for AKI. The role of contrast in CA-AKI is being questioned. In this analysis the only “procedure-related” variable that shows a significant association with AKI is the dose in relation to MACD that by itself is highly correlated with baseline kidney function. An important risk factor for AKI in this population is hemodynamic instability: little data are given on this issue. Is the SBP and HR the value on admission or the worse value “around” the procedure? How many patients had hypotension and/or vasopressor or inotropic support?
Reply # 1
We thank the Reviewer #2 to raise this important issue. We totally agree that hemodynamic status at the time of the procedure is a crucial determinant of the risk of CA-AKI. Regarding SBP and HR values, those reported in the Table are recorded at the beginning of the procedure. To better depict hemodynamic status (as suggested by Reviewer #1), in the current version of the manuscript we added two further variables: hypotension at the time of procedure and implantation of intra-aortic balloon pump. We hope that this data may be of interest for the Reviewer #2.
Modified text: Table 1
Question #2
- an association between AKI and long-term mortality has repeatedly been shown, findings of this analysis are therefore not novel.
Reply # 2
We can understand the skepticism of the Reviewer #2. Similarly, Reviewer #1 emphasized that many studies investigated the prognostic impact of CA-AKI. However, we believe that our study has some peculiarities that should be appreciated:
1) the focus on older patients (aged ≥ 70) all undergoing coronary angiography;
2) all patients received hydration (as correctly underlined by the Reviewer #1);
3) the description of the prognosis in agreement to recovery or not at 3-month of the renal function
In addition, we would like to highlight that we did not confirm in older ACS patients that relationship between CA-AKI and long-term mortality. CA-AKI impacted 3-month death, but only patients that after 3 months did not recover baseline renal function have higher long-term mortality. In older ACS patients, this findings is new and interesting and my help physicians to improve the management of this subgroup of patients.
Question #3
- For me the only conclusion is that patients that develop AKI when hospitalized for ACS have a higher 3-month mortality, but the role of contrast in this is unclear. Whether the association between AKI and long-term mortality is causal cannot be inferred from this type of analyses.
Reply #3
We well understand the issue moved by the Reviewer #2. Accordingly, we damped down the conclusions.
Modified text: section Conclusion, removed text.
Modified text: section Conclusion, page 11, lines 25-29
Despite the use of modern contrast media and hydration protocols, CA-AKI occurs in a relevant portion of older MI patients undergoing invasive strategy and PCI. Patients with CA-AKI showed an increased risk of short-term death. Around two third of the patients with CA-AKI recovered baseline renal function in the first 3 months and their long-term prognosis was similar to that of patients without CA-AKI.
Modified text: section Discussion, page 10, lines 48-50
It is relevant to acknowledge that the number of patients, variables and events of the present study are not enough to clearly demonstrate if CA-AKI is a direct or indirect determinant of death.
Question #4
Other comments
- the timing of CA-AKI is not mentioned – according to KDIGO the 0.3mg/dL increase should occur within 48h – CA-AKI is mostly defined as occurring with 3-5 days
Reply #4
We really thank the Reviewer for this comment, and we are sorry for the imprecision. We systematically applied the definition of CA-AKI based on KDIGO criteria. Then, the timing for the diagnosis occurs in 48-72 hours [22]. This is consistent with many previous papers and now it is clearly stated in the current version of the text.
Modified text: section results, page 3, line 19
CA-AKI was defined based on the Kidney Disease Improving Global Outcomes (KDIGO) criteria: increase in peak serum creatinine of ≥0.3 mg/dL from baseline or ≥1.5 times baseline within 48-72 hours from the invasive procedure.
Added reference number 22:
Chawla LS, Bellomo R, Bihorac A, et al. Acute kidney disease and renal recovery: consensus report of the Acute Disease Quality Initiative (ADQI) 16 Workgroup. Nat Rev Nephrol. 2017 Apr;13(4):241-257.
Question#5
- no data are given on the severity of AKI, that may be an important determinant of “persistent AKI”
Reply #5
The Reviewer #2 is again right. We really appreciated his/her collaboration in improving our paper. Considering the KDIGO definition of stage 3 of AKI (defined as 3-time baseline creatinine, or serum creatinine >4 mg/dl or initial of dialysis), only 6 patients had severe CA-AKI. Of these, 1 patient died in the first 3 months. One patient returns to baseline renal function values. The remaining 4 patients did not recover baseline renal function values. This information is now available in the text.
Modified text: Section Results; page 4, lines 24-26
According to KDIGO definition, CA-AKI was observed in 136 (16%) patients. It was severe (stage 3 of AKI, defined as 3-time baseline creatinine, or serum creatinine>4 mg/dl or initial of dialysis [22]) in 6 patients. Of note, three of them (2.2%) required dialysis during index hospitalization.
Modified text: Section Results; page 8, lines 6-7
Of note, 4 out of the 6 patients with severe CA-AKI in the index hospitalization did not recover baseline renal function.
Question# 6
- the KDIGO definition indeed defines a 0,3 mg/dL increase over 48h as stage 1 AKI. However, I am not sure that this 0,3 mg/dL increase can be used to define “persistent” AKI after 3 months. I would suggest to limit the “persistent” AKI to the fold increase in creatinine.
- in the AKI literature the term persistent AKI is used to define AKI that lasts for more than 48-72h (in contrast to transient AKI). AKI that persists after day 7 is called acute kidney disease, that is defined by fold change in creatinine (Chawla et al, Nature Nephrol 2017). After 3 months kidney function is defined by the CKD criteria. With regard to long-term kidney function the most important analysis would be the change in CKD status between admission and 3 months.
Reply #6
The Reviewer is right. As defined by Consensus statement [Ref 21] persistent AKI is characterized by the continuance of AKI by serum creatinine or urine output criteria (as defined by KDIGO) beyond 48 h from AKI onset. Complete reversal of AKI by KDIGO criteria within 48 h of AKI onset characterizes rapid reversal of AKI. We used the definition of persistent and transient AKI based on the worthy paper of Wi et al [ Ref. 25]. This definition has already been used in the literature and seems to be quite intuitive and effective for Cardiologists. However, as underlined by the Consensus statement [Ref 21], usually the term transient CA-AKI is used to define an episode of CA-AKI that solves in 48-72 hours, if the episode lasts more is called persistent, but after the 7th day it is more correct to talk about acute kidney disease (AKD), even if AKI and AKD are a continuum. Of course, the reason for the lasting of kidney injury are several and the exposition to contrast media might be only an initiator. With the use of the term persistent CA-AKI, we wanted to focus the attention of the reader on this particular point. Thus, apologizing for the confusion, we redefined the terminology in the text. The transient CA-AKI group is now called “recovered renal function group”, whereas the persistent CA-AKI group is now called “not recovered renal function group”. We better define it in methods and among all the paper.
Finally, we do not have data about renal function at three months of all patients enrolled but only of those with CA-AKI, thus we cannot perform the analysis regarding the change in CKD status between admission and 3 months.
Modified text: section Methods, page 3, lines 20-23
If the 3-month value returned to baseline value (according KDIGO criteria) the patient was classified as recovered renal function. On the contrary, if the values persisted to be altered (according KDIGO criteria) the patients was classified as not recovered renal function [22].
Modified text: section Results, page 8, line 1
Three-month recovery of renal functioned in patients with CA-AKI
Modified text: section Results, page 8, line 4, line 5
Overall, 78 (63%) patients showed partial or complete recovery of renal function to the baseline level within 3 months (recovered renal function group). Renal function remained impaired in 45 (37%) patients (not recovered renal function group).
Modified text: section Results, page 8, lines 13-17
In particular, 96 (13%) patients belonged to the no CA-AKI group, 17 (22%) to the recovered renal function group and 21 (47%) to the not recovered renal function group (p<0.001). Unadjusted cumulative mortality beyond 3 months was higher in patients with not recovered renal function (p=0.008 vs. recovered renal function group and p<0.001 vs. no CA-AKI group) (Figure 1).
Modified text: Figure 1
Modified text: section Results, page 8, lines 22-25
After correction for potential confounding factors, the difference was not significant (HR 1.07, 95%CI 0.61-1.91, p=0.751) and patients of the recovered renal function group were not associated with an higher risk of mortality after the 3-month timepoint.
Modified text: Abstract
After 3 months, renal function returned to the baseline value in 78 (63%) with CA-AKI. Those without recovered renal function (n=45, 37%) showed an increased risk of mortality as compared to recovered renal function and no CA-AKI subgroups (HR 2.01, 95%CI 1.55-2.59, p=0.009 and HR 2.71, 95%CI 1.45-5.89, p<0.001, respectively). In conclusion, CA-AKI occurs in a not negligible portion of older MI patients undergoing invasive strategy and it is associated with short-term mortality.
Modified text: Section Discussion, page 11; lines 6-8
However, our analysis did not allow to discriminate from the beginning who will have recovered or not recovered CA-AKI. Then, waiting for future studies able to clarify this issue, physicians can only pursue any effort to avoid CA-AKI.
Modified text: Section Discussion, page 11; lines 13-15
Third, we re-assessed renal function at a single time point (3 months after acute event) and only in patients suffering from CA-AKI in the index hospitalization.
Question #7
- is contrast the cause of “persistent” AKI? No data are available on events that may have caused AKI between the period of CA-AKI and 3 months (eg. how many patients were hospitalized or received RAAS blockers post-discharge)? How many patients without CA-AKI had “AKI criteria” at 3 months?
Reply #7
We agree with the Reviewer. While the analysis of the predictors for CA-AKI is solid and consistent in literature, the one of the predictors of persistent CA-AKI (now renamed in not recovered renal function) is debatable. As suggested by reviewer, many confounders can be involved and hard to be identified. For this reason, we removed the analysis from the manuscript. Of note, the other required information (RAAS blockers treatment) has been added in the manuscript and we would like to highlight that it is not related to not recovered renal function.
Modified text: Table 1
Question #8
Statistics
- in the light of the “rule of ten” the n of variables in the logistic regression analyses is too high with respect to the n of events, especially for the 3 months mortality analysis where only 26 events were noted
Reply to question #8
We thank again Reviewer #2 to raise this important issue. We would like to emphasize that we have 26 deaths in the first 3-months and other 134 deaths after the 3-month timepoint. Overall, we count 160 deaths that should be considered enough for our multivariable model. However, we can understand the criticism of the analysis regarding 3-month mortality. To overcome this limitation, we repeated the Cox regression analysis. We set the inclusion in the multivariable model for variables with p value <0.05 (and not <0.1) at univariate analysis. Furthermore, to avoid overfitting, we applied a backward stepwise modelling approach. We agree that in the current version our findings are more solid. Of note, the findings of the current analysis are totally consistent with the previous ones and CA-AKI remained independent predictor of 3-month mortality.
Modified text: section Methods, page 4, lines 7-13
All variables included in Table 1 were tested using univariate Cox regression as predictors of adverse events. Variables showing a p-value <0.05 were included in a multivariable model in association with CA-AKI with backward stepwise modelling approach. Variables remaining significant with a threshold p-value ≤0.05 were retained as final predictors. Cox regression analyses were performed for 3-month mortality and for mortality beyond the 3-month time point.
Modified text: section Results, page 6, lines 9-11
At multivariable analysis, after correction for potential confounding factors, CA-AKI emerged as independent predictor of 3-month death (HR 3.51, 95%CI 1.05-7.01) (Table 2).
Question #10
- were variables checked for collinearity (that probably exists between eGFR and MACD)
Reply to question #10
We thank again the Reviewer #2 for his/her constructive comment and support. We really apologize for not being sufficiently clear about this point. We agree with Reviewer that MACD and baseline creatinine clearance have related each other. The computation of MACD value starts from baseline creatinine clearance. However, neither MACD nor baseline CrCl emerged in the current multivariable model. For these considerations, we believe that our model is solid and well describes the outcome of patients and the independent predictors.

Round 2
Reviewer 1 Report
I have no additional comments
Author Response
Reviewer #1
Question #1
I have no additional comments
Reply #1
Thank you very much for the time spent on our paper and for your important suggestions.
Reviewer 2 Report
In this secondary analysis of two observational trials the authors assess the impact of contrast-associated AKI on long-term adverse outcomes. They report that CA-AKI is associated with increased 3months mortality and if persistent at 3 months also with long-term mortality.
General comment
In general the focus of this paper is too much on contrast. This is actually a paper on “AKI and mortality in patients admitted with ACS and requiring angiography”.
This applies to:
- the development of AKI where the only contrast-related independent predictor of CA-AKI relates to the MACD that is highly correlated with baseline kidney function (the Mehran score is not procedure-related but reflects the patient’s risk factors).
- the (non)recovery of CA-AKI since there are no data on kidney function nor on events with risk for AKI in the period between the procedure and 3 months (not even a kidney function at hospital discharge, which is strange since that is the moment of inclusion of the original trials).
- the impact of CA-AKI on mortality: this is the impact of contrast “associated”, not contrast-induced AKI, thus including all the AKI risk factors in the early phase of ACS and in the period thereafter. The association of AKI and its (non)recovery with mortality is generally known.
- the methods section where “procedures for CA-AKI prevention” only mentions contrast and the discussion that focusses on hydration and contrast dose
- An observational study cannot infer causality and thus wording such as “impact” and “determinant of death” should be avoided. This analysis can only show associations and cannot “clarify the missing pieces of the puzzle”. Note: it is not the “number of patients, variables and events” that is the problem but the design of the study.
The analysis
- the authors now mention the number of patients with AKI stage 3. What was the number with stage 2 and how did this affect non-recovery? This could be very interesting information for the clinician with regard to follow-up of kidney function
- with regard to the n of variables: the rule of 10 applies to the number of variables that are considered for the analysis (here all the variables in table 1) and not the number of variables that eventually are included in the multivariate model. This means that instead of 35 variables you theoretically can only consider 3 variables for 3mth mortality and 14 for log-term mortality and you therefore have a very high risk of overfitting.
- I still have doubts about the definition of (non)recovery. The 0.3mg/dL increase that is part of the KDIGO definition is supposed to occur within 48h and should be interpreted in the clinical context. In addition, changes in serum creatinine of this magnitude may be part of the “normal” fluctuations, especially in patients with reduced eGFR where a small change in GFR will result in a larger increase in Screat (Newhouse et al. Am J Rontgenol 2008; 191: 376-82)
Author Response
Reviewer # 2
Question#1
In this secondary analysis of two observational trials the authors assess the impact of contrast-associated AKI on long-term adverse outcomes. They report that CA-AKI is associated with increased 3-month mortality and if persistent at 3 months also with long-term mortality.
General comment
In general the focus of this paper is too much on contrast. This is actually a paper on “AKI and mortality in patients admitted with ACS and requiring angiography”.
This applies to:
- the development of AKI where the only contrast-related independent predictor of CA-AKI relates to the MACD that is highly correlated with baseline kidney function (the Mehran score is not procedure-related but reflects the patient’s risk factors).
- the (non)recovery of CA-AKI since there are no data on kidney function nor on events with risk for AKI in the period between the procedure and 3 months (not even a kidney function at hospital discharge, which is strange since that is the moment of inclusion of the original trials).
- the impact of CA-AKI on mortality: this is the impact of contrast “associated”, not contrast-induced AKI, thus including all the AKI risk factors in the early phase of ACS and in the period thereafter. The association of AKI and its (non)recovery with mortality is generally known.
- the methods section where “procedures for CA-AKI prevention” only mentions contrast and the discussion that focusses on hydration and contrast dose.
- An observational study cannot infer causality and thus wording such as “impact” and “determinant of death” should be avoided. This analysis can only show associations and cannot “clarify the missing pieces of the puzzle”. Note: it is not the “number of patients, variables and events” that is the problem but the design of the study.
Reply # 1
We are sorry for being not enough clear. In relation to each single point raised in the general comment:
- We agree with the Reviewer and we clearly stated in discussion that CA-AKI is an entity highly debated. In our paper we discussed about “contrast associated” and not “contrast-induced” to further support the temporal association between contrast exposure and AKI; being aware that other important factors play a crucial role. In highly selected population of older MI patients undergoing invasive treatment we found that MACD play an important role.
- We are sorry for the misinterpretation. In the database we have data about the creatine values of patients at discharge, but based on previous studies [e.g. ref 23] we design our analysis on three month time-point to better distinguish short term from long term impact.
- We completely agree with the Reviewer and for this reason we distinguish the short term outcome from long term outcome. The long term outcome is determined not only by contrast exposure (that could be an initiator) but from the persistence of renal disfunction (as described in consensus paper [ref. 22]).
- We agree with the Reviewer, as suggested this is a study on exposure on contrast in patients undergoing coronary angiography and sub-sequent development of AKI. In the paragraph “procedures for CA-AKI prevention” we summarize what is per Guidelines suggested to prevent CA-AKI and then we discussed it.
- We are sorry for the imprecision, as suggested we amended the sentence in discussion.
Modified text: Section Discussion, page 10, lines 22-24.
It is relevant to acknowledge that the design of the study (observational) is not enough to infer causality and to clearly demonstrate if CA-AKI is a direct or indirect determinant of death.
Question # 2
The analysis
- the authors now mention the number of patients with AKI stage 3. What was the number with stage 2 and how did this affect non-recovery? This could be very interesting information for the clinician with regard to follow-up of kidney function.
Reply # 2
We thank the Reviewer for the comment. Patients with moderate AKI were 17. Of these three did not recover baseline renal function at the 3-month follow-up.
Modified text: section Results, page 4, lines 13-18
It was severe (stage 3 of AKI, defined as 3-time baseline creatinine, or serum creatinine>4 mg/dl or initial of dialysis [22]) in 6 patients and moderate in 17 patients (stage 2 of AKI, defined as 2 to 2.9-time baseline creatinine [22]). Of note, three of patients with severe AKI (2.2%) required dialysis during index hospitalization.
Modified text: section Results, page 7, lines 36-38
Of note, 4 out of the 6 patients with severe CA-AKI and 3 out of 17 with moderate CA-AKI in the index hospitalization did not recover baseline renal function.
Question# 3
- with regard to the n of variables: the rule of 10 applies to the number of variables that are considered for the analysis (here all the variables in table 1) and not the number of variables that eventually are included in the multivariate model. This means that instead of 35 variables you theoretically can only consider 3 variables for 3mth mortality and 14 for log-term mortality and you therefore have a very high risk of overfitting.
Reply # 3
We are sorry for being not enough clear. Firstly, rule of ten is very unstable and it has been largely debated in literature and it is not the aim of this study to validate or not it [Journal of clinical epidemiology 2015: 68; 627-636]. Secondly, we would like to stress that in the second version of the analysis to avoid overfitting we applied two corrections:
- including in multivariable model only variables with a p value <0.05 and not <0.1
- a backward stepwise modelling.
These are methods usually accepted to avoid overfitting and confirmed our preliminary results, suggesting the reliability of our analysis.
Question # 4
- I still have doubts about the definition of (non)recovery. The 0.3mg/dL increase that is part of the KDIGO definition is supposed to occur within 48h and should be interpreted in the clinical context. In addition, changes in serum creatinine of this magnitude may be part of the “normal” fluctuations, especially in patients with reduced eGFR where a small change in GFR will result in a larger increase in Screat (Newhouse et al. Am J Rontgenol 2008; 191: 376-82)
Reply # 4
We are sorry for being not enough clear. We applied the KDIGO definition for CA-AKI. For being not misinterpreted with the definition of persistent and transient CA-AKI, we decided as explained in the last revision to talk about “recovered” and “not-recovered renal function” at three month. Of course as already explained factors occurring in worsening of renal failure are several. One of the initiator of AKI could be the exposition to contrast. Some patients recovered and some other not also because of other variables intervening. Anyway as defined in consensus document “Persistent acute kidney injury (AKI) is characterized by the continuance of AKI by serum creatinine or urine output criteria (as defined by KDIGO) beyond 48 h from AKI onset. (consensus statement 1A);… AKI and acute kidney disease (AKD) are a continuum, and persistent AKI frequently becomes AKD, defined as a condition wherein criteria for AKI stage 1 or greater persists ≥7 days after an exposure (consensus statement 1C)… Acute kidney disease (AKD) describes acute or subacute damage and/or loss of kidney function for a duration of between 7 and 90 days after exposure to an acute kidney injury (AKI) initiating event (consensus statement 2)”. Thus considering the consensus document we think that our definition might be used to evaluate a group of patient with an episode of AKI after contrast exposure, that even in presence of other contributing causes, did not recover renal function and showing that this is related to worse prognosis.
